# EV Cargo Sorting in Therapeutic Development for Cardiovascular Disease

**DOI:** 10.3390/cells10061500

**Published:** 2021-06-15

**Authors:** Cherrie D. Sherman, Shweta Lodha, Susmita Sahoo

**Affiliations:** Cardiovascular Research Center, Icahn School of Medicine at Mount Sinai, One Gustav L. Levy, P.O. Box 1030, New York, NY 10029, USA; cherrie.sherman@mssm.edu (C.D.S.); shweta.lodha@mssm.edu (S.L.)

**Keywords:** cardiovascular disease, extracellular vesicles, exosomes, therapeutics, diagnostics, cargo sorting pathogenesis

## Abstract

Cardiovascular disease remains the leading cause of morbidity and mortality in the world. Thus, therapeutic interventions to circumvent this growing burden are of utmost importance. Extracellular vesicles (EVs) actively secreted by most living cells, play a key role in paracrine and endocrine intercellular communication via exchange of biological molecules. As the content of secreted EVs reflect the physiology and pathology of the cell of their origin, EVs play a significant role in cellular homeostasis, disease pathogenesis and diagnostics. Moreover, EVs are gaining popularity in clinics as therapeutic and drug delivery vehicles, transferring bioactive molecules such as proteins, genes, miRNAs and other therapeutic agents to target cells to treat diseases and deter disease progression. Despite our limited but growing knowledge of EV biology, it is imperative to understand the complex mechanisms of EV cargo sorting in pursuit of designing next generation EV-based therapeutic delivery systems. In this review, we highlight the mechanisms of EV cargo sorting and methods of EV bioengineering and discuss engineered EVs as a potential therapeutic delivery system to treat cardiovascular disease.

## 1. Introduction

Intercellular communication, an essential facet of cellular function, is achieved through a highly organized and unique system involving signaling molecules. Most cell types orchestrate a highly coordinated packaging of biochemical signals in membrane-bound vesicles called extracellular vesicles (EVs) that are ubiquitously secreted to extracellular milieu [1,2,3,4]. Once thought of as ‘garbage bags’ or ‘cellular dust’, EVs are now known to play a major role in angiogenesis, immune regulation, cellular growth, development, differentiation, migration and other physiological processes [5,6,7,8].

The emerging roles of EVs in intercellular communication, disease pathogenesis and diagnosis have increased interest in investigating their potential to serve as a therapeutic vehicle [1,5,9,10,11]. EVs as a molecular delivery system has become a topic of intense research due to their innate ability to carry a repertoire of proteins, nucleic acids, lipids, metabolites, virus particles, or organelles from parent cells and modulate the function of recipient cells [1,6,9,10,12,13]. The cargo from EVs has a distinct molecular fingerprint reflecting the originating cells that facilitates selective incorporation and triggers biochemical and/or phenotypic changes in the recipient cells [2,5,13,14].

EVs show promise to serve as a biomarker and treatment delivery tool for cardiovascular disease (CVD), a class of disorders pertaining to malfunctions of the heart and blood vessels [15]. For example, patients at risk of or with ischemic or nonischemic coronary artery disease show significantly altered levels of EVs secreted by certain cell types [16]. In another study, EVs engineered to contain cardioprotective miRNAs reduced myocardial damage and infarction in rats [17].

Despite accrual of scientific evidence highlighting the function of EVs and their therapeutic potential, we do not have a comprehensive understanding of the mechanisms of their origin and secretion. Here, in this review, we discuss EV biology with a special focus on cargo sorting and their cellular uptake, which will aid in designing superior EV-based therapeutic vehicles to treat CVD. We will also discuss current EV engineering methods employed to deliver therapeutic molecules, and the applications of this technology in CVD treatment.

## 2. Classification of Extracellular Vesicles

Currently, a gold standard EV nomenclature has not been set despite collective experimental effort to understand EV biology although this area is actively evolving. Lack of standardization of EV isolation techniques, coupled with EV heterogeneity, poses a significant challenge to EV isolation and subsequent characterization [18,19,20]. For instance, the accuracy of conventional tools used to measure the size of isolated particles, such as flow cytometry, dynamic light scattering, nanoparticle tracking analysis, and more has shown to be hindered by low refractive index, size heterogeneity, lack of adequate measurement range and variable sample handling prior to measurement [20,21]. To underscore these limitations, a guideline called minimal information for studies of extracellular vesicles (MISEV) was created by the International Society for Extracellular Vesicles (ISEV) for EV studies (for more info, see [22]). Indeed, various techniques such as identification of molecular biomarkers or signatures and development of better isolation protocols aid in EV classification and characterization.

According to MISEV, EVs are a generic term for particles naturally released from the cell that are delimited by a lipid bilayer and cannot replicate, i.e., do not contain a functional nucleus [22]. EVs are loosely grouped in to different categories, exosomes, microvesicles (MV) and apoptotic bodies, based on their size and biogenesis [2,5]. Exosomes are known to be a heterogenous population, containing both large and small vesicles ranging from ~30 to 120 nm, with endosomal origin [23]. Interestingly, there is a recently discovered abundant population of non-membranous nanoparticles known as exomeres (~35 nm) [23]. MVs are larger (~100 nm and above) plasma membrane-derived ectosomes [22].

Apoptotic bodies originate during programmed cell death, or apoptosis, and are also large vesicles (~50–5000 nm) [1,2,5,10,11]. Unlike exosomes and MVs, which are secreted during normal cellular activity, apoptotic bodies are only formed during apoptosis [24]. Apoptotic bodies may originate intra- or extracellularly, and are typically characterized by the presence of organelles within vesicles [24]. Interestingly, smaller apoptotic bodies (~50–500 nm) resembling MVs or exosomes can be secreted during apoptosis [24]. While there is some evidence to suggest that membrane blebbing plays a role in the release of smaller apoptotic bodies, the origin of these vesicles remains understudied [24,25]. Moreover, though limited findings suggest that apoptotic bodies may transfer genetic and protein content intercellularly, further work is needed to shed light on the functional and therapeutic potential of such vesicles [24,26,27,28]. Due to a lack of conclusive literature explaining the origin and function of apoptotic bodies, in this review, we focus on exosomes and MVs as potential therapeutic vehicles to treat CVD.

### Cargo Sorting in Extracellular Vesicles

Cells’ ability to secrete, recognize and ingest bioactive EVs signifies supreme cargo sorting and delivery mechanisms. Deciphering the molecular mechanism of cargo sorting would increase our ability to manipulate and utilize EVs to carry and deliver therapeutic molecules to target cells.

The luminal cargo of EVs mimics their cellular origin and includes genes actively expressed at the time of packaging and/or release [2,29,30]. For instance, EVs derived from cancer cells, virus- infected cells, stem cells and myocytes carry distinctly different cargo [5,11,31,32,33]. Interestingly, the same cells can excrete morphologically different EVs with varying nucleic acid, lipid and protein content [2,10,31]. Conditions such as stress stimuli can also affect EVs content. For example, under hypoxic conditions, cardiomyocytes secrete exosomes containing apoptotic and autophagic molecules such as tumor necrosis factor-α (TNF-α) and miRNA-30a [34,35]. Moreover, the analysis of cargo content may be affected by the techniques utilized to isolate and prepare the EVs [36,37,38]. For instance, a recent study showed that exosome populations isolated from ultracentrifugation, total exosome precipitation, OptiPrep density gradient and size exclusion chromatography had significantly different glycosylation profiles [37]. Additionally, commonly used isolation methods can cause isolated vesicles to become contaminated with non-EV bioactive factors, and subsequently cause misattribution of observed cellular effects to the cargo content of EVs [39].

A comprehensive study analyzing the proteomic and lipidomic qualities of exosomes and MVs secreted by different cell types (U87 glioblastoma cells, Huh7 hepatocellular carcinoma cells and human bone marrow-derived mesenchymal stem cells (MSCs)) shows diversity in cargo composition [31]. In this study, Haraszti and colleagues reported that the protein and lipid composition of exosomes and MVs differed despite a shared cellular origin. In addition, the proteome of exosomes from cancer cell types was similar but different from that of stem cell-derived exosomes. These findings exemplify that cells use distinct mechanisms in EV cargo sorting. Furthermore, differences in cargo content are cell-dependent, which highlights the specific nature of communication between the donor and recipient cells. Databases of proteins, lipids and RNAs detected in EVs from different cell sources are found in (http://www.exocarta.org) ExoCarta [40] and Vesiclepedia [41].

Exosomes expressing global markers such as tumor susceptibility gene 101 (TSG101), ALG-2-interacting protein X (ALIX) or heat shock protein 70 (HSP70) indicate that intraluminal vesicles (ILV) were formed via the endosomal sorting complexes required for transport (ESCRT)-dependent pathway while those that contain tetraspanins CD81, CD9 and CD63 were formed via the ESCRT independent pathway [42,43,44,45,46,47,48]. It is plausible that ESCRT-dependent and -independent pathways work in synergy, which may explain exosomes’ heterogeneous cargo composition irrespective of a shared cellular origin.

## 3. Exosomal Biogenesis and Cargo Sorting

Highly regulated cargo sorting in exosomes involves a multi-layer of complex instructions. Matrix-bound nanovesicles (MBVs) or late endosomes start from early (or sorting) endosomes (EEs), which form from fusion of endocytic vesicles [49,50]. Stringent cargo sorting begins in EEs while patrolling the peripheral cytoplasm near the plasma membrane (PM). Incoming endocytosed cargo (e.g., receptors and ligands) that fuse with EE are either retained, recycled back to the PM or destined for complete or partial degradation [49,50,51]. EE’s architectural structure includes a vacuolar space where endocytosed cargo is sorted, and extending tubules that are involved in molecular sorting, generation of recycling endosomes and vesicle targeting to specific organelles such as the PM and the trans-golgi network [49,50].

EE cargo sorting and recycling relies on endosomal acidification. For example, low-density lipoprotein and transferrin receptors are recycled back to the PM upon releasing their ligands at pH ~6.5, while signaling receptors may remain bound to their ligand at pH ~4.5. pH-mediated cargo sorting and recycling within EEs continues until selected cargo are sorted into ILVs [51]. In addition, accumulating evidence indicates that cargo recycling is dependent on the ability for cargo sorting machinery to recognize and associate with specific short-sequence motifs (~4–7 amino acids) of cargo molecules. These sequence motifs, existing within the carboxyl-terminus domain of membrane receptors and proteins, are critical for internalization, cargo sorting and trafficking in subcellular compartments [52,53].

For more in-depth reading of small peptide recognition sequences for intracellular sorting, we recommend these review articles (References [53,54]). Other important molecular identifiers required in cargo sorting are protein post-translational modifications (PTMs). These versatile regulatory processes modulate protein conformation, stability and function, and in the context of EVs, regulate cargo sorting, secretion and uptake [48,55,56,57,58,59,60,61,62,63].

### 3.1. Protein Sorting in ILVs: ESCRT-Dependent and -Independent Mechanisms

Exosomes are the extracellular form of ILVs, which are generated within the multivesicular body (MVB) via inward budding of the MVB or late endosomal membrane [2,3,64,65]. ILVs discriminately capture specific molecules which may be incorporated as part of the membrane or cargo. Exosomes, regardless of their cellular origin, contain endosome-associate proteins such as Rab GTPase, SNAREs (soluble NSF attachment protein receptors), annexins and flotillin which are involved in membrane fusion, transport of vesicles and MVB biogenesis [3,66]. MVBs can either fuse with lysosomes to degrade their content or release ILVs to extracellular space [3,10]. Exosome biogenesis and cargo sorting rely on two intricate (ESCRT)-dependent or ESCRT-independent mechanisms that direct the fate of exosomes or ILVs in MVBs (Figure 1).

#### 3.1.1. The Role of ESCRT in Exosome Biogenesis

ESCRT has four protein complexes termed ESCRT–0, –I, –II and –III, assembled from ~20 proteins and their cognate proteins VPS4, VTA1 and ALIX. ESCRT complexes play a crucial role in cargo selection, endosome formation, exosome release and MVB degradation [3,10,42,43].

ILV biogenesis begins in EEs, which then fuse with late endosomes [33]. ESCRT controls the sorting of ubiquitinated proteins into ILVs via recognition of ubiquitin molecule (~8.5 kDa) tagged in the protein’s lysine residue/s. Ubiquitination is a reversible process involved in endocytic trafficking, signal transduction, inflammation, DNA repair, translation, and protein degradation [48,55,67].

#### 3.1.2. ESCRT Components HRS, STAM1/2, TGS101 and CHMP

On the endosomal membrane, ESCRT-0 and its cognate proteins, hepatocyte growth factor-regulated tyrosine kinase substrate (HRS) and STAM 1/2 (signaling transducing adaptor molecules 1 and 2) subunits harbor ubiquitin-binding domains that recognize ubiquitinated proteins. ESCRT-I subunit TSG101, which recognizes and binds to ubiquitinated proteins, is recruited by HRS [48,49,50,55,63]. TSG101 also binds to the small integral membrane protein of the lysosome/late endosome (SIMPLE), motifs responsible for endocytic activities. Conditional knock-out of HRS and mutation in SIMPLE motifs reduces exosome secretion and affects epidermal growth factor (EGF)**^+^**–MVB formation [68,69]. Interestingly, loss of HRS, TGS101, VPS22 and VPS24 affects MVB biogenesis but does not totally abolish MBV production, which implies that another pathway mediates exosome biogenesis. Other ESCRT complexes such as ESCRT–I and –II complexes regulate the initial invagination process of the endosomal membrane, while ESCRT–II and ESCRT–III with CHMP (chromatin modifying protein) 2, 3 and 4 proteins further promote vesicles’ budding and scission [42,43].

#### 3.1.3. Deubiquitination and Other Post-Translational Modifications of Exosomal Proteins

Deubiquitination of K63- (a ubiquitin molecule involved in different cellular processes) and K48-linked (a ubiquitin degradation signal) proteins by ESCRT-III-recruited deubiquitinase precedes sorting into ILVs. Deubiquitinases such as associated molecule with the SH3 domain of STAM (AMSH)-ubiquitin-isopeptidase can recognize and deubiquitinate proteins with K63-linked chains and ubiquitin-isopeptidase-Y (UBPY) can remove both K63- and K48-linked chains from proteins. In contrast, some proteins do not undergo this type of post-translational modification prior to cargo loading into ILVs (e.g., MHCII in DC). Studies have detected the presence of ubiquitinated proteins in ILVs bound for MVBs’ degradation (e.g., ubiquitinated MHCII), which suggests a sorting mechanism independent of ESCRT complexes [48,56]. Other PTMs detected in exosomal proteins include phosphorylation, sumoylation, glycosylation, oxidation, isgylation, myristoylation, prenylation, myristoylation, citrullation and others (reviewed in References [48,61,70,71]).

#### 3.1.4. ESCRT-Independent Exosome Biogenesis

In the absence of all ESCRT complexes, exosome biogenesis depends on lipids such as ceramide, lysobisphosphatidic acid (LBPA) and tetraspanins (CD63, CD9, CD81, CD82, and CD151) that are crucial for membrane remodeling, budding and protection from hydrolases in the extracellular environment [29,44,45,46,47,72]. Tetraspanins are exosomal markers, and regulate actin polymerization, membrane budding and EV cargo loading [46]. ALIX requires LBPA to recruit ESCRT–III proteins to endosomes to assist in sorting and incorporation of tetraspanins to ILVs, and syndecans and syntenin to promote intraluminal budding and abscission [73,74]. Interaction of proteins (e.g., metalloproteinase CD10, Epstein-Barr virus (EBV) protein LMP1 during EBV replication and premelanosome *protein* (*PMEL*) during melanogenesis) with the cytoplasmic domain of tetraspanins results in their incorporation into ILVs [63]. Tetraspanins CD82 and CD9 also associate with cytoplasmic proteins E-cadherin and β-catenin which alter the Wnt signaling pathway in the recipient cells [75].

#### 3.1.5. Role of Sphingolipid Ceramide in ESCRT-Independent Exosome Biogenesis

Sphingolipid ceramide, on the other hand, triggers budding of exosomes into MBVs and promotes maturation of exosomes within MVBs when hydrolyzed to sphingosine 1-phosphate (S1P) by sphingomyelinase [45,47]. Sphingosine kinase 2 (Sphk2) or S1P1 receptors are required in the formation of CD63, CD81 and flotillin-positive exosomes [47]. Exosomes enriched in ceramides and cholesterol rely on sphingomyelinase 2 (nSMase2) for ceramide production and ceramides for their delivery to the recipient cells. Other exosomes such as amyloid-β-peptide^+^-exosomes are dependent on sphingomyelin synthase SMS2 for their secretion [48]. During MVB formation, incorporation of cytosolic proteins in a microautophagy-like mechanism destined for complete or partial degradation requires the chaperon heat shock cognate 71 kDa (HSC70), and ESCRT-I and -III [76].

### 3.2. RNA Sorting in ILVs

#### 3.2.1. Role of PTMs in RNA Sorting

Small and long non-coding RNAs, microRNAs (miRNAs), mRNAs, siRNA and structural RNAs have been detected in EVs [3,77]. Of these different RNA species, miRNAs have attracted the most attention, due to their regulatory roles in gene expression. There are conflicting reports of (i) the abundance of EV-associated and non-EV associated miRNAs in bodily fluids, (ii) the physiological relevance of EV-associated miRNAs and (iii) the abundance of miRNAs as an RNA species in EVs. Volumes of EV literature suggests that miRNAs are one of the most abundant RNA species present in EVs [78,79,80].

Interestingly, there are a few reports that contradict this concept. In a well-controlled study, Tewari and group suggested that EV miRNAs may not be a major contributor of miRNAs identified as biomarkers within biofluids [81]. Along this line, another manuscript recently submitted to bioRxiv proposes that EV-associated miRNAs are only a minor constituent of EV RNA content, and that very little miRNAs present in EVs are delivered to the target cells [82]. However, this article did not provide significant direct evidence to support their claims. A key factor impacting the conclusions drawn from these studies are the methods used to isolate the vesicles [78]. We have yet to address questions like whether ultracentrifugation-based isolation can enrich particular EVs with specific types of RNA species. Nevertheless, it is widely established that multiple different types of RNA species are carried by EVs, and that they are actively or passively sorted in to EV compartments [78,79,80].

To date, the exact mechanism behind RNA sorting still remains unclear. Studies unraveling RNA sorting mechanisms show that sumoylation is a PTM signature detected in miRNAs packaged in EVs. miRNAs from B-cells and urine show PTMs including adenylation and uridylation at the their 3’ end [59]. miRNAs are ~22 nucleotide non-coding RNA molecules that regulate gene expression. The RNA binding protein heterogeneous nuclear ribonucleoprotein (hnRNP) plays a critical role in sorting miRNA and long non-coding RNAs (lncRNAs) into exosomes. Sumoylated hnRNPA2B1 and hnRNPA1 selectively sort miRNAs via a CCGA or CCCU and GAGAG recognition signal in their 3’ end respectively [59,83]. In cancer-derived EVs, Lupus La selectively sorts miRNA-122 via preferential recognition of its two motifs [72]. In chronic inflammation, the EXOmotif in miRNA-939 is involved in exosome loading [84].

#### 3.2.2. Roles of KRAS, RISC and Dicer in miRNA Sorting

Inhibition of nSMase2, which produces ceramide, leads to a decrease in the number of miRNA molecules in EVs. In contrast, nSMase2 inhibition leads to an increase in the amount of miR-100 level in KRAS mutant cells, which implicates the role of small GTPase, KRAS, in this cell type. KRAS mutation also affects intraluminal vesiculation of several miRNAs such as the oncogenic miR-10b in colorectal cancer cells [85]. A few proteins involved in the maturation of miRNA and regulation of miRNA activity have been implicated in mediating miRNA loading. Small RNAs direct RNA-induced silencing complexes (RISCs) to mediate miRNA gene silencing activity, and are physically and functionally associated with MVBs and miRNA sorting. Transfection of pre-miRNAs in exosomes and their maturation into miRNAs are mediated by Dicer [86,87].

Dicer and RISC components, Ago2 and GW182 have been detected in exosomes [86,88,89]. Presence of Ago2 in exosomes and its regulatory role in exosomal miRNA sorting is controversial. Many studies have shown expression of Ago2 within exosomes derived from human cancer cells, HEK293 cells and human pericardial fluid [84,86,90]. For instance, one study showed that Ago2 secretion within exosomes derived from isogenic colorectal cancer cells is regulated by KRAS activation and subsequent MEK-ERK signaling [91]. McKenzie et al. suggested that while Ago2 is an exosomal cargo of most cell types, low levels may be detected due to signaling regulation [91]. Moreover, a knockdown of argonaute 2 (Ago2) has been shown to decrease the level of miR-451, miR-150 and miR-142-3p in EVs produced by HEK293 cells in vitro [86,87]. In contrast, in human circulating plasma, Ago2 is shown to be primarily associated with non-vesicular protein-RNA complexes [92]. Furthermore, exosomes secreted from monocytes human contained only small amounts of Ago2 [88]. It is not clear whether Ago-2 secretion and association with EV-miRNAs is cell and species-specific and a thorough investigation is needed to determine the role of Ago2 in miRNA secretion and sorting within exosomes.

#### 3.2.3. Role of RNA Modification in EV Cargo Sorting

Recent studies by Dr. Schekman and colleagues have shown that tRNAs in smaller EVs such as exosomes undergo exosome-specific modifications and are more enriched in exosomes compared to cells [93]. There remain unidentified additional or alternative modifications at the dihydrouridine sites that must be characterized. Nevertheless, these and other RNA modifications essential under physiological and pathological conditions may implicate a potential role for RNA modifications in RNA cargo sorting within EVs [94]. Moreover, a related study showed YBX1, a known RNA binding protein secreted by exosomes, to play a role in sorting miRNAs into the intralumenal vesicles of multivesicular bodies for export via unconventional secretion mechanisms in HEK293T cells [95]. Physiological studies of the function of miRNAs secreted via Ago2-associated vs. chaperone-mediated pathways may explain contradictory results for different miRNAs and provide general rules for extracellular miRNA function.

## 4. Microvesicles Biogenesis and Cargo Sorting

Microvesicles (also known as ectosomes, oncosomes, shedding vesicles, shedding bodies and microparticles) are formed via direct budding from the PM, which releases their content into extracellular space [29,96,97]. Various mRNAs, miRNAs, lncRNAs and proteins have been consistently detected in MVs [3,31,96,98,99]. Their biogenesis begins with reception to signaling molecules such as growth factors and cytokines while their cargo sorting is dependent on PM oligomerization [70]. Shen and colleagues have shown that PM anchors (e.g., N-terminal acylation tag, myristoylation tag, phosphatidylinositol-(4,5)-bisphosphate (PIP2)-binding domain, phosphatidylinositol-(3,4,5)-trisphosphate-binding domain, prenylation/palmitoylation tag and a type-1 plasma membrane protein, CD43) can target cytoplasmic proteins (e.g., oligomeric cytoplasmic protein, TyA) into MVs and to the budding site [99].

MVs exocytosis relies on calcium- or lipid-mediated mechanisms [29,96,97]. Increase in intracellular calcium is followed by PM and cytoskeletal changes. Calcium-sensitive proteins such as gelsolin and calpain alter MVs actin cytoskeleton, leading to their detachment from PM and initiation of PM blebbing and vesiculation. Modulation of aminophospholipid translocase, flippase and lipid scramblase functions via elevated calcium signaling, which leads to a loss of plasma membrane asymmetry, destabilization of PM-cytoskeleton contact and exposure of phosphatidylserine [29,70,97,100]. MVs also share part of the ESCRT mechanism employed by exosomes. TSG101, a component of ESCRT–I binds to arrestin domain-containing protein 1 (ARRDC1) in the PM to promote MV exocytosis via Gag-mediated budding [29,101].

## 5. Therapeutic Potential oF EVs in Cardiovascular Disease

The utilization of EVs in a clinical setting to treat CVD or block disease progression via delivery of specific drugs, proteins, miRNA and functional genes remains an active topic in cardiovascular research. In particular, attempts to identify cell-derived exosomes carrying cardioprotective molecules shows promising results. In addition, the accumulating evidence on EV cargo sorting mechanisms continues to pave the way for engineering EVs as a therapeutic delivery system [6,102,103,104,105,106,107,108,109].

### 5.1. Transport of Cardioprotective miRNAs via Exosomes

Ischemia and cardiomyocyte necrosis resulting from myocardial infarction requires regenerative rescue. Various studies demonstrated the potential for EVs carrying cardioprotective cargo to improve cardiovascular function, and thus be used as an alternative cell-free therapeutic. For example, exosomes enriched with miR-21-5p and miR-210-3p from mouse cardiac fibroblast-derived induced pluripotent stem cells (iPS cells) prevented apoptosis in H9C2 (embryonic rat cardiomyocytes) via regulation of Nanog and Hypoxia-inducible factor-α (HIF-α) respectively [110]. Similarly, miR-210 derived from cardiac progenitor cells blocks cardiomyocyte apoptosis by downregulating ephrin and PTP1b, while miRNA-132 downregulates RasGAP-p120, which enhances endothelial cell tube formation [111]. Paracrine secretion of exosomes with pro-angiogenic molecules from human CD34^+^ cells exhibit angiogenic activity in ischemic hindlimb [112,113].

#### Hypoxia Can Increase Therapeutic Potential of Exosomes

Various studies have implicated that hypoxia and inflammation may affect cardiovascular function via alteration of EV cargo [34,35,114]. Hypoxia-induced cardiomyocyte death is a contributor to acute myocardial infarction. Numerous studies have shown that cardiomyocytes exposed to hypoxia release EVs with cardioprotective cargo [114]. Hypoxic rat cardiomyoblasts secrete EVs containing cardioprotective miRNAs including miR-21-5p, miR-378-3p, miR-152-3p and let-7i-5p, which promote cell survival. Notably, overexpression of miR-21-5p reduces the expression of tumor suppressor PTEN and programmed cell death 4, and upregulates the anti-apoptotic gene, Bcl-2 expression while overexpression of miR-152-3p and let-7i-5p decreases apoptotic mediators, Atg12 and Faslg mRNA levels [114]. Similarly, exosomes from Akt, HIF-1α, or CXCR4-overpressed MSCs enhance angiogenesis and improve cardiac function [115,116,117]. Similar studies from our lab have shown that hypoxia increases the angiogenic and therapeutic potential of CD34+ stem cell-derived exosomes [118].

### 5.2. EV Modification: Sorting Bioactive Molecules into Exosomes

Studies on the use of exosomes as therapeutics use viral and non-viral methods to modify exosomal cargo and surface molecules for efficient delivery and cellular uptake (Figure 1).

#### 5.2.1. Methods for Engineering Exosomes

Kim and colleagues demonstrated the use of exosomes as a drug delivery platform to treat multiple drug-resistant (MDR) cancers [119]. They developed different loading methods such as room temperature incubation, electroporation and sonication to incorporate a chemotherapeutic agent, paclitaxel (PTX) in exosomes collected from RAW 264.7 macrophages. Exosomes containing PTX (exoPTX) were taken up ~30 times more than synthetic nanocarriers. Furthermore, uptake of exoPTX increased drug cytotoxicity in MDR cancer cells compared to those treated with PTX alone. Most of the cargo loading techniques such as sonication, direct transfection, electroporation and saponin permeabilization were adapted from the liposome delivery approach [120]. A similar approach using an anti-inflammatory compound named curcumin was incorporated in exosomes via incubation (Table 1). This method of manipulating exosomal cargo content was also applied to load an oligonucleotide in exosomes. Didiot and colleagues developed oligonucleotide therapeutics to target *Huntingtin* mRNA and protein by incorporating hydrophobically modified small interfering RNAs (hsiRNAs) into exosomes through incubation [121].

#### 5.2.2. Exosome-Mediated Gene Therapy

The field of gene therapy has boomed since its comeback in the 21st century after many failed attempts in the late 20th century [132]. Recent studies employing the use of viral vectors to replace a non-functional or deleted gene generated promising results. One example is the incorporation of a protein, TNF-related apoptosis-inducing ligand (*TRAIL*) in exosomes using a lentiviral human membrane TRAIL to produce TRAIL^+^ exosomes in K562 cells which suppress tumor progression in vivo and also induce caspase-3-mediated cell death in cancer cells [133]. The use of exosome-associated AAV (adeno-associated virus) to enhance liver gene transfer and evade pre-existing humoral immunity was also successful in mice with hemophilia B [134]. Studies have shown that multiple viruses use the ESCRT machinery to produce enveloped virions which avoid immune surveillance [109]. Our lab is currently exploring the potential for exosome-associated AAV containing a gene with cardioprotective and regenerative properties to treat CVD [126,135].

#### 5.2.3. Target Specificity of Therapeutic Txosomes

Production of therapeutic EVs requires signaling biomolecules that are specific to target cells for precise delivery. Hung and Leonard reported that the peptides fused to the N-terminus of Lamp2b, a transmembrane protein on exosomal surface, are proteolytically cleaved during exosome biogenesis [136]. To circumvent this event, they engineered peptide-Lamp2b fusion protein with a glycosylation motif at various sites. This modification stabilized the peptide, and subsequently enhanced exosomal targeted trafficking to the recipient cells. Similarly, a novel targeting system was generated using Lamp2b fused to a cardiomyocyte-specific peptide, WLSEAGPVVTVRALRGTGSW, which resulted in increased uptake and retention by cardiomyocytes compared to that seen in non-targeted exosomes [137]. A summary of the studies employing exosomes containing cardioprotective and bioactive molecules with or without modification, their target molecules and downstream potential to ameliorate different cardiovascular conditions are listed in Table 1.

## 6. EVs as Biomarkers

EVs are present in biological fluids such as blood, urine, saliva, seminal and amniotic fluid, milk, bile, semen and malignant ascites [1,3,5,6,10]. Their presence in most body fluids makes them a robust potential biomarker for CVD and other diseases [16].

### 6.1. Microvesicles as Biomarkers

High-risk patients for CVD and acute coronary syndrome show increased levels of endothelial and platelet-secreted microvesicles [16,138]. Familial hypercholesterolemia has also been associated with more platelet-derived and tissue factor-rich microvesicles [16,139]. The unique content of cell-derived EVs also holds diagnostic and prognostic potential. For instance, a study evaluating 181 patients with clinically stable CAD (coronary artery disease) over the course of 6 years revealed that increased miR-126 and miR-199a in circulating MVs is associated with a lower risk of future major adverse cardiac event [140].

### 6.2. Exosomes as Biomarkers

Exosomes have shown comparatively less potential to serve as an efficient biomarker, due in part to more complex exosome isolation and quantification protocols [16]. Due to their small size, exosomes are often undetected by commonly used devices such as flow cytometers, and subsequently require specialized equipment for quantification, including nanoparticles tracking analysis (NTA) monitors or dynamic light scattering (DLS) machines [141]. Despite these constraints, there exist limited findings revealing the predictive power of exosomes in CVD and other diseases. For instance, exosomes from patients with acute coronary syndrome and myocardial infarction showed higher levels of miR-1 and miR-133a [142,143]. Furthermore, plasma from ovarian cancer patients contained a significantly higher number of exosomes than that from healthy individuals [144,145]. Taken together, these findings reveal the potential for exosomes to serve as biomarkers and subsequent need for simpler exosome extraction and quantification methods.

## 7. Challenges in the Clinical Application of EVs

### 7.1. Therapeutic Limitations of EVs

While the use of engineered vesicles containing therapeutic agents to treat CVD and other diseases is very promising, effective targeting and uptake of EVs as a therapeutic delivery tool remains a challenging feat. When engineering therapeutic EVs, several complex factors must be taken into consideration, including identification of specific EV-producing cells, replication of optimal vesicle production conditions and manipulation of vesicle cargo content.

The clinical application of EVs is also limited by physiological barriers preventing the uptake of therapeutic miRNAs by mammalian cells [146]. For instance, orally ingested RNAs may become degraded by lytic enzymes secreted by the saliva, or denatured within the acidic environment of the stomach, which can prevent efficient cellular uptake of therapeutic miRNAs by the host system [146]. Moreover, efficient and targeted trafficking and cellular uptake of EVs by cardiovascular tissues requires robust investigation to avoid delivery to and uptake by non-target healthy cells. Finally, pre-existing diseases (cancer, viral infection, autoimmune), stress, genetic aberration and humoral immunity may alter the uptake and function of EVs due to ongoing pathophysiological and biochemical changes in the recipient cells.

EV therapeutic strategies require extensive optimization and standardization prior to clinical application [103]. Upscaling EV production poses a challenge for therapeutic use as cells produce low number of exosomes. Nasiri Kenari and colleagues developed a method to produce artificial EVs as an alternative to endogenous EVs. These biomimetic EVs (also known as exosome-mimetics and mimetic nanovesicles) bypass endosomal sorting and exhibit characteristics such as size, morphology and membrane protein markers similar to conventional EVs [147,148,149]. Despite these promising results, further studies are required to overcome the challenges preventing efficient use of EVs to treat CVD.

### 7.2. Challenges of Using EVs as Biomarkers

Lack of standardization of sample collection, isolation and quantification currently deter the potential use of EVs in diagnostics and prognostics. Substantial evidence highlights the impacts of disparate sample handling on the molecular and physical properties of EVs [150,151]. In addition, when working with protein-rich biological serum or plasma, it can be difficult to purify EVs from non-vesicular protein components [152]. The level of circulating EVs are also known to be influenced by the time of day when the sample is collected and amount of physical activity undertaken prior to collection [151].

Moreover, it remains challenging to isolate tissue-specific EVs from the human body. EVs isolated from bodily fluids display greater heterogeneity than those from cell culture supernatant [153,154]. For instance, human plasma contains mixtures of exosomes subsets secreted by diverse cell types [153,154,155]. Moreover, physiological fluids contain EVs secreted from both diseased and non-diseased cells, which complicates the recovery of disease-specific EVs as biomarkers [153,154,155]. Furthermore, lack of capture reagents with sufficient specificity for specific exosome markers exacerbates the challenge of isolating pure EVs from human plasma [153].

## 8. Conclusions

EVs present a promising therapeutic strategy for CVD due to their biocompatibility, paracrine and endocrine presence, low immunogenicity, low toxicity and ability to carry bioactive molecules to target cells. Studies on cargo content, cargo sorting, signaling peptides and the development of EV engineering methods have provided a general view on how to manipulate nanovesicles to generate an effective therapeutic delivery system and biomarker for CVD. However, challenges in isolating, purifying, and manipulating EVs must be resolved to optimize the translation of EVs in a clinical setting.

## Figures and Tables

**Figure 1 cells-10-01500-f001:**
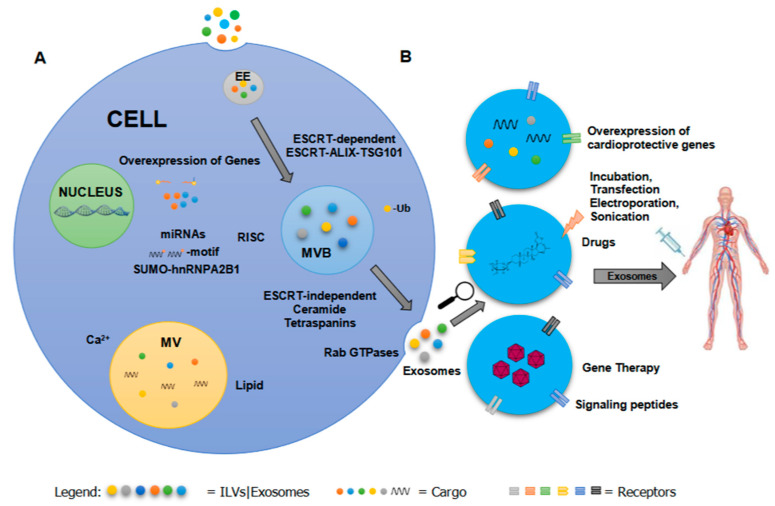
A summary of (**A**) EV cargo sorting and (**B**) methods for engineering exosome-based therapeutics to treat cardiovascular disease. EV cargo sorting is dependent on recognition signal or motif, post-translational modifications, and signaling molecules via ESCRT-dependent or -independent mechanism. Current methods in engineering EVs to incorporate therapeutic molecules include viral (e.g., overexpression of a gene using a viral vector, and delivery of a gene using a recombinant virus) or non-viral (e.g., incorporation of drugs or bioactive compounds by incubation or mechanical means, and alteration of targeting peptides) approaches. Abbreviations: Extracellular Vesicle (EV), Endosomal Sorting Complex (ESCRT), Early Endosome (EE), Multivesicle (MV), Multivesicular body (MVB).

**Table 1 cells-10-01500-t001:** Studies employing exosomes with or without modification for treatment of cardiovascular disease (CVD). Up- or down-regulation of target molecules is symbolized by red or blue arrows respectively.

EV Sources	Therapeutic Agents	Target Molecules	Mode of Action	Species/Diseases Conditions	References
Exosomes from cardiac fibroblast-derived induced pluripotent stem cells (iPS cells)	miR-21-5pmiR-210-3p	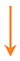	ROSFLASHCasp8ap2	Anti-apoptoticCell survival	Mouse/ Myocardial Ischemia and Reperfusion	[110]
Exosomes from cardiomyocytes	miR-222miR-143	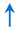	CD31+ cells	Improved neovascularizatio n	Mouse/Acute Myocardial Infarction	[122]
Exosomes from cardiac progenitor cells	miR-210 miR-132	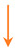	EphrinPTP1bRasGAP-p120	Anti-apoptoticCell survivalEnhance endothelial tube formation	Mouse/Myocardial Infarction	[111]
Exosomes from hypoxic rat cardiomyoblasts	miR-21-5pmiR-378-3p miR-152-3plet-7i-5p	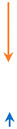	PTENPDCD4Atg12FaslgBcl-2	Anti-apoptoticCell survival	Acute Myocardial Infarction	[114]
Exosomes from human pericardial fluid	miR-let-7b-5p	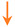	TGFBR1	Vascular remodelingPro-angiogenic	Mouse/Limb Ischemia	[90]
Exosomes from human pericardial fluid	Clusterin	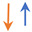	EMT genes	Epicardial ActivationArteriogenesisAnti-apoptotic	Mouse/Acute Myocardial Infarction	[123]
Exosomes from CD34^+^ stem cells	miR-126-3pShh	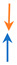	SPRED1PTCH1GLI TFs	Vascular remodelingPro-angiogenic	Mouse/Myocardial Ischemia	[113] [124]
Exosomes from rat plasma	HSP70	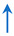	TLR4ERK1/2p38MAPK	Anti-apoptotic	Rat/Ischemic-Reperfusion Injury	[125]
AAV-mediatedGene therapy	SERCA2a	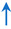	Intracellular Ca^2+^	Enhanced cardiomyocyte contractility	Mouse/Chronic Heart Failure	[9,126]
Exosomes from CXCR4- overexpressing lentiviral transduced MSC	CXCR4	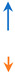	IGF-1αpAkTCaspase 3	Cardiac remodelingPro-angiogenic	Rat/Myocardial Infarction	[115]
Exosomes from Akt-overexpressing adenoviral transduced MSC	Akt	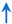	PDGF-D	Vascular endothelial formationPro-angiogenic	Rat/Acute Myocardial Infarction	[116]
Exosomes from HIF-1α-overexpressing lentiviral transduced MSC	HIF-1α	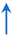	Jagged1Notch target genes	Endothelial formationPro-angiogenic	Mouse/Myocardial Ischemia	[117]
Curcumin in exosomes from EL-4 cells	Curcumin (a bioactive compound)	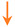	TGFβ1MMP-1, -9	Inhibits myofibroblast differentiationPromote collagen degradation	Mouse/Septic Shock	[127,128]

Note: This table focuses on key miRNAs and proteins with therapeutic potential to treat CVD. Existing literature shows miRNAs to be one of the key functional content of EVs. However, EVs are shown to contain many other RNA species, including protein-coding mRNAs (or their fragments), mtRNAs (mitochondrial RNA), snoRNA (small nucleolar RNA), yRNA, piRNA (piwi-interacting RNA), lncRNA (long non-coding RNA) and vRNA (vault RNA), and a diverse array of proteins that could drive phenotypic differences [129,130]. For instance, Lopatina et al. showed that EVs secreted by platelet-derived growth factors can protect tissue from ischemic injury by transporting and inducing the expression of lncRNA MALAT1 (Metastasis Associated Lung Adenocarcinoma Transcript 1), a well-known pro-angiogenic and anti-inflammatory regulator [131].

## Data Availability

No new data were created in this study. Data sharing is not appli-cable to this article. This manuscript adheres to the MDPI Research Data Policies at https://www.mdpi.com/ethics.

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
