# Peer review of "EV Cargo Sorting in Therapeutic Development for Cardiovascular Disease"

_cells, 2021, doi:10.3390/cells10061500_

Round 1
Reviewer 1 Report
Dr. Sherman in the paper submitted to Cells (cells: 968645
Title: EV Cargo Sorting in Therapeutic Development for 2 Cardiovascular Disease) made a review the mechanisms of EV cargo sorting and methods of EV bioengineering and discuss engineered EVs as a potential therapeutic delivery system to treat cardiovascular disease. The review is well written, rich of important information well explained. I think that a little part on apoptotic bodies should be added just to frame the argument on EV, even if few are known on this latter.
I believe that final consideration should be longer than now explained better the point-of-view of the author.

Author Response
*The reviewer’s comments are in black and italicized, and our response is in blue.
Dr. Sherman in the paper submitted to Cells (cells: 968645 Title: EV Cargo Sorting in Therapeutic Development for 2 Cardiovascular Disease) made a review the mechanisms of EV cargo sorting and methods of EV bioengineering and discuss engineered EVs as a potential therapeutic delivery system to treat cardiovascular disease. The review is well written, rich of important information well explained. I think that a little part on apoptotic bodies should be added just to frame the argument on EV, even if few are known on this latter. I believe that final consideration should be longer than now explained better the point-of-view of the author.
We thank the reviewer for their positive comments. To accommodate their suggestion, we have expanded our discussion of apoptotic bodies in Section 2: Classification of EVs. We have specifically added the following, underlined below, which also encompasses Line 74-85 of our revised manuscript:
Apoptotic bodies originate during programmed cell death, or apoptosis, and are also large vesicles (~50-5000 nm) [1-5].Unlike exosomes and MVs, which are secreted during normal cellular activity, apoptotic bodies are only formed during apoptosis [6]. Apoptotic bodies may originate intra- or extracellularly, and are typically characterized by the presence of organelles within vesicles [6]. Interestingly, smaller apoptotic bodies (~50-500 nm) resembling MVs or exosomes can be secreted during apoptosis [6]. While there is some evidence to suggest that membrane blebbing plays a role in the release of smaller apoptotic bodies, the origin of these vesicles remains understudied [6, 7]. Moreover, though limited findings suggest that apoptotic bodies may transfer genetic and protein content intercellularly, further work is needed to shed light on the functional and therapeutic potential of such vesicles [6, 8-10]. Due to a lack of conclusive literature explaining the origin and function of apoptotic bodies, in this review, we focus on exosomes and MVs as potential therapeutic vehicles to treat CVD.
Added References:
- Tancini, B., et al., Insight into the Role of Extracellular Vesicles in Lysosomal Storage Disorders. Genes (Basel), 2019. 10(7).
- Colombo, M., G. Raposo, and C. Théry, Biogenesis, secretion, and intercellular interactions of exosomes and other extracellular vesicles. Annu Rev Cell Dev Biol, 2014. 30: p. 255-89.
- Maas, S.L.N., X.O. Breakefield, and A.M. Weaver, Extracellular Vesicles: Unique Intercellular Delivery Vehicles. Trends Cell Biol, 2017. 27(3): p. 172-188.
- Willms, E., et al., Extracellular Vesicle Heterogeneity: Subpopulations, Isolation Techniques, and Diverse Functions in Cancer Progression. Front Immunol, 2018. 9: p. 738.
- Mihaela Gherghiceanu, N.A., Stefania Lucia Magda, Alina Constantin, Miruna Nemecz, Alexandru Filippi, Octavian Costin Ioghen, Laura Cristina Ceafalan, Florina Bojin, Gabriela Tanko, Virgil Paunescu, Dragos Vinereanu, Ewa Stepien and Adriana Georgescu, https://www.intechopen.com/books/extracellular-vesicles-and-their-importance-in-human-health/part-one-extracellular-vesicles-as-valuable-players-in-diabetic-cardiovascular-diseases, in Extracellular Vesicles and Their Importance in Human Health, J.A.R.-C. Ana Gil De Bona, Editor. 2019. p. 1-25.
- Akers, J.C., et al., Biogenesis of extracellular vesicles (EV): exosomes, microvesicles, retrovirus-like vesicles, and apoptotic bodies. J Neurooncol, 2013. 113(1): p. 1-11.
- Coleman, M.L., et al., Membrane blebbing during apoptosis results from caspase-mediated activation of ROCK I. Nat Cell Biol, 2001. 3(4): p. 339-45.
- Samos, J., et al., Circulating nucleic acids in plasma/serum and tumor progression: are apoptotic bodies involved? An experimental study in a rat cancer model. Ann N Y Acad Sci, 2006. 1075: p. 165-73.
- Liu, D., et al., Circulating apoptotic bodies maintain mesenchymal stem cell homeostasis and ameliorate osteopenia via transferring multiple cellular factors. Cell Res, 2018. 28(9): p. 918-933.
- Bergsmedh, A., et al., Horizontal transfer of oncogenes by uptake of apoptotic bodies. Proc Natl Acad Sci U S A, 2001. 98(11): p. 6407-11.
Reviewer 2 Report
This is a review of “EV Cargo Sorting in Therapeutic Development for Cardiovascular Disease” by Sherman et al. In this manuscript the authors do a deep dive on EVs covering a number of topics related to cargo sorting in EVs (exosomes and MVs) as well as their potential roles in therapeutics and as biomarkers. Overall it is a well-written piece with useful information on EV formation and cargo loading into these EVs. I enjoyed reading it and think it will be a useful addition to the literature. However, my enthusiasm for the piece was tempered a bit by the lack of critical analysis or presenting the other side of some of the controversial science. The Sahoo lab is very strong and I think this manuscript can be improved by wading deftly into these controversial areas.
- Although the authors discuss MISEV, it would be useful for some statements about how different EV prep methods can result in drastically different results and thus many of the papers described might not replicate well depending on the exact methods used. This also pertains to isolation methods as your line 74-75 talks about different cargo, which is correct, but might be somewhat dependent on isolation methodology. As well, this recent paper in JEV was particularly interesting for suggesting we misattribute cellular effects to EVs, when that is not the case. https://www.tandfonline.com/doi/full/10.1080/20013078.2020.1807674.
- Somewhat relative to that last point above is your citation 82. This well-cited paper focused on differences in two miRNAs of iPSC-exos and CF-exos to claim the effect. However, I imagine there are other miRNAs, proteins, small molecules, tRFs, etc that are different between those exosomes that could just as likely drive phenotype differences. This paper and other like it in the manuscript is where you could provide some critical analysis throughout the piece. As an example, I was impressed with the way controversy was handled in this recent review of the even more controversial xenomiRs: https://peerj.com/articles/9567/.
- miRNAs in EVs is controversial, going back to the Tewari paper https://www.ncbi.nlm.nih.gov/pmc/articles/PMC4205618/ and more recently https://www.biorxiv.org/content/10.1101/2020.05.20.106393v2. As well, this paper by Tosar https://academic.oup.com/nar/article/43/11/5601/1172894 suggests there is no specific sorting. The two issues are 1) how much miRNA material is in EVs and 2) are they actively sorted. This controversy should be covered rather than the way the paper is written suggesting that all cargo is sorted.
- Under 7.2 (line 371) you need to also discuss, as a limitation, how circulating EVs are really a whole-body biopsy, unless one is able to sub-select EVs coming from a specific cell type due to a unique surface marker. So, if one is interested in neural EVs, they would likely represent a small fraction of all EVs in blood, and small (but biologically important) changes in cargo might be diluted out. This is another current limitation of EVs as biomarkers for specific diseases.
Minor issues.
- Missing a verb in the sentence on line 267.
- Line 355 needs a space between tool/remains.
Author Response
*Reviewer’s comments are in black and italicized, and our response is in blue.
This is a review of “EV Cargo Sorting in Therapeutic Development for Cardiovascular Disease” by Sherman et al. In this manuscript the authors do a deep dive on EVs covering a number of topics related to cargo sorting in EVs (exosomes and MVs) as well as their potential roles in therapeutics and as biomarkers. Overall it is a well-written piece with useful information on EV formation and cargo loading into these EVs. I enjoyed reading it and think it will be a useful addition to the literature. However, my enthusiasm for the piece was tempered a bit by the lack of critical analysis or presenting the other side of some of the controversial science. The Sahoo lab is very strong and I think this manuscript can be improved by wading deftly into these controversial areas.
- Although the authors discuss MISEV, it would be useful for some statements about how different EV prep methods can result in drastically different results and thus many of the papers described might not replicate well depending on the exact methods used. This also pertains to isolation methods as your line 74-75 talks about different cargo, which is correct, but might be somewhat dependent on isolation methodology. As well, this recent paper in JEV was particularly interesting for suggesting we misattribute cellular effects to EVs, when that is not the case. https://www.tandfonline.com/doi/full/10.1080/20013078.2020.1807674.
We thank the reviewer for their extremely helpful and intriguing insights. In response to the reviewer’s first point, we have expanded Section 2: Classification of EVs, and Section 2.1: Cargo Sorting in EVs of our manuscript to better highlight the controversies underlying EV characterization and isolation.
In Section 2: Classification of EVs, we have specifically added the following, encompassing Line 55-63 of our revised manuscript:
Lack of standardization of EV isolation techniques, coupled with EV heterogeneity, poses a significant challenge to EV isolation and subsequent characterization [1-3]. For instance, the accuracy of conventional tools used to measure the size of isolated particles, such as flow cytometry, dynamic light scattering, nanoparticle tracking analysis, and more has shown to be hindered by low refractive index, size heterogeneity, lack of adequate measurement range, and variable sample handling prior to measurement [3, 4]. To underscore these limitations, a guideline called minimal information for studies of extracellular vesicles (MISEV) was created by the International Society for Extracellular Vesicles (ISEV) for EV studies (for more info, see [5]).
In Section 2.1: Cargo Sorting in EVs, we have added the following, encompassing Line 96-102 of our revised manuscript. We have specifically included the interesting reference suggested by the reviewer in Line 100-102:
Moreover, the analysis of cargo content may be affected by the techniques utilized to isolate and prepare the EVs [6-8]. For instance, a recent study showed that exosome populations isolated from ultracentrifugation, total exosome precipitation, OptiPrep density gradient, and size exclusion chromatography had significantly different glycosylation profiles [7]. Additionally, commonly used isolation methods can cause isolated vesicles to become contaminated with non-EV bioactive factors, and subsequently cause misattribution of observed cellular effects to the cargo content of EVs [9].
- Somewhat relative to that last point above is your citation 82. This well-cited paper focused on differences in two miRNAs of iPSC-exos and CF-exos to claim the effect. However, I imagine there are other miRNAs, proteins, small molecules, tRFs, etc that are different between those exosomes that could just as likely drive phenotype differences. This paper and other like it in the manuscript is where you could provide some critical analysis throughout the piece. As an example, I was impressed with the way controversy was handled in this recent review of the even more controversial xenomiRs: https://peerj.com/articles/9567/
In response to the reviewer’s suggestion #2, we have added a thorough paragraph detailing the limitations of our Table 1, which is described by Lines 307- 313 of our revised manuscript, and states the following:
Note: This table focuses on key miRNAs and proteins with therapeutic potential to treat CVD. Existing literature shows miRNAs to be one of the key functional content of EVs. However, EVs are shown to contain many other RNA species, including protein-coding mRNAs (or their fragments), mtRNAs, snoRNA, yRNA, piRNA, lncRNA, and vRNA, and a diverse array of proteins that could drive phenotypic differences [10, 11]. For instance, Lopatina et al. showed that EVs secreted by platelet derived growth factors can protect tissue from ischemic injury by transporting and inducing the expression of lncRNA MALAT1, a well-known pro-angiogenic and anti-inflammatory regulator [12].
Moreover, we found the paper about xenomiRs to be intriguing, and expanded Section 7.1: Therapeutic Limitations of EVs to include it, described below and within Lines 417-421 of our revised manuscript:
The clinical application of EVs is also limited by physiological barriers preventing the uptake of therapeutic miRNAs by mammalian cells [13]. For instance, orally ingested RNAs may become degraded by lytic enzymes secreted by the saliva, or denatured within the acidic environment of the stomach, which can prevent efficient cellular uptake of therapeutic miRNAs by the host system [13].
- miRNAs in EVs is controversial, going back to the Tewari paper https://www.ncbi.nlm.nih.gov/pmc/articles/PMC4205618/ and more recently https://www.biorxiv.org/content/10.1101/2020.05.20.106393v2. As well, this paper by Tosar https://academic.oup.com/nar/article/43/11/5601/1172894 suggests there is no specific sorting. The two issues are 1) how much miRNA material is in EVs and 2) are they actively sorted. This controversy should be covered rather than the way the paper is written suggesting that all cargo is sorted.
In line with the reviewer’s suggestions, we have expanded Section 3.2.1: Role of PTMs in RNA sorting, listed below and as Lines 222-238 of our newly revised manuscript.
Of these different RNA species, miRNAs have attracted the most attention, due to their regulatory roles in gene expression. There are conflicting reports of i) the abundance of EV-associated and non-EV associated miRNAs in bodily fluids, ii) the physiological relevance of EV-associated miRNAs and iii) the abundance of miRNAs as an RNA species in EVs. Volumes of EV literature suggests that miRNAs are one of the most abundant RNA species present in EVs [14-16]. However, there are a few reports that contradict this concept. In a well-controlled study, Tewari and group suggested that EV miRNAs may not be a major contributor of miRNAs identified as biomarkers within biofluids [17]. Along this line, another manuscript recently submitted to bioRxiv proposes that EV-associated miRNAs are only a minor constituent of EV RNA content, and that very little miRNAs present in EVs are delivered to the target cells [18]. However, this article did not provide significant direct evidence to support their claims.
A key factor impacting the conclusions drawn from these studies are the methods used to isolate the vesicles [14]. We have yet to address questions like whether ultracentrifugation-based isolation can enrich particular EVs with specific types of RNA species. Nevertheless, it is widely established that multiple different types of RNA species are carried by EVs, and that they are actively or passively sorted in to EV compartments [14-16].
Furthermore, we have also inserted a small paragraph on the exciting field of RNA modifications and its potential role in RNA cargo sorting under Section 3.2 RNA sorting in ILVs. We have specifically included the following section as Lines 274-285 of our revised manuscript:
3.2.3 RNA modification in EV cargo sorting
Recent studies by Dr. Schekman and colleagues have shown that tRNAs in smaller EVs such as exosomes undergo exosome-specific modifications and are more enriched in exosomes compared to cells [19]. There remain unidentified additional or alternative modifications at the dihydrouridine sites that must be characterized. Nevertheless, these and other RNA modifications essential under physiological and pathological conditions may implicate a potential role for RNA modifications in RNA cargo sorting within EVs [20]. Moreover, a related study showed YBX1, a known RNA binding protein secreted by exosomes, to play a role in sorting miRNAs into the intralumenal vesicles of multivesicular bodies for export via unconventional secretion mechanisms in HEK293T cells [21]. Physiological studies of the function of miRNAs secreted via Ago2-associated vs. chaperone-mediated pathways may explain contradictory results for different miRNAs and provide general rules for extracellular miRNA function.
- Under 7.2 (line 371) you need to also discuss, as a limitation, how circulating EVs are really a whole-body biopsy, unless one is able to sub-select EVs coming from a specific cell type due to a unique surface marker. So, if one is interested in neural EVs, they would likely represent a small fraction of all EVs in blood, and small (but biologically important) changes in cargo might be diluted out. This is another current limitation of EVs as biomarkers for specific diseases.
To address the reviewer’s suggestion, we have expanded Section 7.2: Challenges of using EVs as biomarkers.We have specifically added the following, also included as Lines 442-448 in our revised manuscript:
Moreover, it remains challenging to isolate tissue-specific EVs from the human body. EVs isolated from bodily fluids display greater heterogeneity than those from cell culture supernatant [22, 23]. For instance, human plasma contains mixtures of exosomes subsets secreted by diverse cell types [22-24]. Moreover, physiological fluids contain EVs secreted from both disease-and non-diseased cells, which complicates the recovery of disease-specific EVs as biomarkers [22-24]. Furthermore, lack of capture reagents with sufficient specificity for specific exosome markers exacerbates the challenge of isolating pure EVs from human plasma [22].
Minor issues.
- Missing a verb in the sentence on line 267.
- Line 355 needs a space between tool/remains
We have addressed the minor issues pointed out by the reviewer. In our revised manuscript, you may find the corrects as respectively Lines 326 and 413-414.
Added References:
- Tauro, B.J., et al., Comparison of ultracentrifugation, density gradient separation, and immunoaffinity capture methods for isolating human colon cancer cell line LIM1863-derived exosomes. Methods, 2012. 56(2): p. 293-304.
- Nolan, J.P. and J. Moore, Extracellular vesicles: Great potential, many challenges. Cytometry B Clin Cytom, 2016. 90(4): p. 324-5.
- Witwer, K.W., et al., Standardization of sample collection, isolation and analysis methods in extracellular vesicle research. J Extracell Vesicles, 2013. 2.
- Erdbrugger, U. and J. Lannigan, Analytical challenges of extracellular vesicle detection: A comparison of different techniques. Cytometry A, 2016. 89(2): p. 123-34.
- Théry, C., et al., Minimal information for studies of extracellular vesicles 2018 (MISEV2018): a position statement of the International Society for Extracellular Vesicles and update of the MISEV2014 guidelines. J Extracell Vesicles, 2018. 7(1): p. 1535750.
- Tang, Y.T., et al., Comparison of isolation methods of exosomes and exosomal RNA from cell culture medium and serum. Int J Mol Med, 2017. 40(3): p. 834-844.
- Freitas, D., et al., Different isolation approaches lead to diverse glycosylated extracellular vesicle populations. J Extracell Vesicles, 2019. 8(1): p. 1621131.
- Taylor, D.D. and S. Shah, Methods of isolating extracellular vesicles impact down-stream analyses of their cargoes. Methods, 2015. 87: p. 3-10.
- Whittaker, T.E., et al., Experimental artefacts can lead to misattribution of bioactivity from soluble mesenchymal stem cell paracrine factors to extracellular vesicles. J Extracell Vesicles, 2020. 9(1): p. 1807674.
- van Balkom, B.W., et al., Quantitative and qualitative analysis of small RNAs in human endothelial cells and exosomes provides insights into localized RNA processing, degradation and sorting. J Extracell Vesicles, 2015. 4: p. 26760.
- Abramowicz, A. and M.D. Story, The Long and Short of It: The Emerging Roles of Non-Coding RNA in Small Extracellular Vesicles. Cancers (Basel), 2020. 12(6).
- Lopatina, T., et al., PDGF enhances the protective effect of adipose stem cell-derived extracellular vesicles in a model of acute hindlimb ischemia. Sci Rep, 2018. 8(1): p. 17458.
- Mar-Aguilar, F., et al., Evidence of transfer of miRNAs from the diet to the blood still inconclusive.PeerJ, 2020. 8: p. e9567.
- Murillo, O.D., et al., exRNA Atlas Analysis Reveals Distinct Extracellular RNA Cargo Types and Their Carriers Present across Human Biofluids. Cell, 2019. 177(2): p. 463-477 e15.
- Gallo, A., et al., The majority of microRNAs detectable in serum and saliva is concentrated in exosomes. PLoS One, 2012. 7(3): p. e30679.
- Cheng, L., et al., Exosomes provide a protective and enriched source of miRNA for biomarker profiling compared to intracellular and cell-free blood. J Extracell Vesicles, 2014. 3.
- Chevillet, J.R., et al., Quantitative and stoichiometric analysis of the microRNA content of exosomes.Proc Natl Acad Sci U S A, 2014. 111(41): p. 14888-93.
- Albanese, M., et al., Micro RNAs are minor constituents of extracellular vesicles and are hardly delivered to target cells. bioRxiv, 2020: p. 2020.05.20.106393.
- Shurtleff, M.J., et al., Broad role for YBX1 in defining the small noncoding RNA composition of exosomes. Proc Natl Acad Sci U S A, 2017. 114(43): p. E8987-E8995.
- Mathiyalagan, P., et al., FTO-Dependent N(6)-Methyladenosine Regulates Cardiac Function During Remodeling and Repair. Circulation, 2019. 139(4): p. 518-532.
- Shurtleff, M.J., et al., Y-box protein 1 is required to sort microRNAs into exosomes in cells and in a cell-free reaction. Elife, 2016. 5.
- Whiteside, T.L., Extracellular vesicles isolation and their biomarker potential: are we ready for testing?Ann Transl Med, 2017. 5(3): p. 54.
- Hong, C.S., et al., Isolation of biologically active and morphologically intact exosomes from plasma of patients with cancer. J Extracell Vesicles, 2016. 5: p. 29289.
- Yekula, A., et al., From laboratory to clinic: Translation of extracellular vesicle based cancer biomarkers. Methods, 2020. 177: p. 58-66.